# Extreme waves analysis based on atmospheric patterns classification: an application along the Italian coast

Francesco De Leo[1], Sebastián Solari[2], and Giovanni Besio[1]

[1]Dept. of Civil, Chemical and Environmental Engineering - University of Genoa, Genoa, 16145, Italy
[2]Instituto de Mecánica de los Fluidos e Ingenieria Ambiental - Universidad de la República, Montevideo, 11300, Uruguay

**Correspondence:** Francesco De Leo (francesco.deleo@edu.unige.it)

**Abstract.**

This work provides a methodology for classifying samples of significant wave height peaks in subsets homogeneous in terms of the atmospheric circulation patterns behind the observed extreme wave conditions. Then, a methodology is given for the computation of the overall extreme value distribution starting from the distributions fitted to each single subset. To this end, *k-means* clustering technique is used to classify the shape of the wind fields that occurred simultaneously and prior to the occurrence of the extreme wave events. This results in a small number of characteristic circulation patterns, each one associated to a subset of extreme wave values. After fitting an extreme value distribution to each subset, bootstrapping is used to reconstruct the *omni*-circulation pattern extreme value distribution.

The methodology is applied to several locations among the Italian buoy network, and from the obtained results it is concluded that it yields a two-fold advantage: first, is capable of identifying clearly differentiated subsets driven by homogeneous circulation patterns; second, it allows to estimate high return-period quantiles consistent with those resulting from the usual extreme value analysis. In particular, the circulation patterns highlighted are analyzed in the context of the Mediterranean sea atmospheric climatology, and shown to be due to well-known cyclonic systems typically crossing the Mediterranean basin.

## 1 Introduction

The extreme value theory is widely used for the analysis of extreme data in most of the geophysical applications. It allows to estimate extreme (un-observed) values, starting from available records or modelled data which are assumed to be independent and identically distributed (Coles, 2001). It is therefore crucial to identify homogeneous datasets complying with the above mentioned requirement before performing the EVA of a given physical quantity.

When dealing with directional variables, it is common to group the data according to different directional sectors (Cook and Miller, 1999; Forristall, 2004), being such approach recommended as well in many regulations (API, 2002; ISO, 2005; DNV, 2010, among others). However, the use of directional sectors involves certain drawbacks. First, it cannot be employed for variables being not characterized by incoming directions (such as storm surge or rainfall). Second, data showing the same direction may be due to different forcing; in the frame of wave climate, an example is that of waves propagating in shallow waters, affected by refraction and/or diffraction. Finally, the borders of the directional sectors are often subjectively defined,

without verifying if the data belonging to each subset are homogeneous and independent with respect to those of the other sectors (see Folgueras et al., 2019, where they tackled this issue and proposed a methodology to overcome it).

An alternative approach to classify the extremes implies resorting to the atmospheric circulation conditions they are driven by, associating each extreme event to a particular weather pattern (referred to as WP). Such approach has been already deepseated in atmospheric sciences for the analysis of precipitations, snowfalls, temperature, air quality and winds (Yarnal et al.,

2001; Huth et al., 2008, among others). Nevertheless, there are few studies linking weather circulation patterns with the most likely induced sea states (e.g., wave climate and storm surge). Holt (1999) classified WPs leading to extreme storm surges in the Irish Sea and the North Sea. Guanche et al. (2013) simulated multivariate hourly sea state time series in a location in the northwester Spanish coast, starting from the simulation of weather pattern time series. Dangendorf et al. (2013) linked the atmospheric pressure fields with the sea level in the German Bight (southeastern North Sea). Pringle et al. (2014, 2015)

investigated how extreme wave events may be tied to synoptic-scale circulation patterns in the east coast of South Africa. Camus et al. (2014) proposed a statistical downscaling of sea states based on weather types, then applied to a couple of locations in the Atlantic coast of Europe to hindcast the wave climate during the twentieth century plus modelling it under different climate change scenarios. The latter methodology was improved by Camus et al. (2016), and further used by Rueda et al. (2016) for analysing significant wave height maxima. Solari and Alonso (2017) used WP classification to perform EVA

of significant wave heights in the south-east coast of South America.

Except for Rueda et al. (2016) and Solari and Alonso (2017), none of the previous works focused on exploiting WP classification methodologies for defining homogeneous datasets to be further employed for EVA. However, the two methodologies differ in several aspects. Rueda et al. (2016) dealt with daily maxima significant wave heights along with surface pressure fields and pressure gradients, averaging over different time periods and applying a regression-guided classification to define

100 WPs. They subsequently fit a Generalized Extreme Value distribution, estimating an Extremal Index from the daily maxima significant wave height of each WP, from which they rebuilt the overall distribution of annual maxima (referred to as AM $H_s$). Despite the proposed methodology is able to reproduce the AM $H_s$ distribution, with such a large number of WPs it may be difficult to detect the most relevant physical processes behind the occurrence of extreme wave conditions. Furthermore, as shown in Rueda et al. (2016), even though a large number of WPs was considered, only a few happened to significantly affect

the EVA, as most of the WPs resulted to be associated to mild wave conditions. Finally, to retain daily maxima does not ensure the data to be independent, thus implying the need to use the Extremal Index. Solari and Alonso (2017) introduced instead a "bottom-up" scheme: they first selected a series of independent extreme sea states; then, they identified a reduced number of WPs that allows to group the selected data into homogeneous populations. A small number of WPs makes it easier to link the different subsets of extremes with known climate forcing. Above all, to work with independent peaks allows to rely on the

classic and well known extreme value theory, with no need to refer to additional indexes and/or more complex models that may be unfamiliar for many analysts.

In this paper, the methodology of Solari and Alonso (2017) is revisited and applied to several wave datasets along the Italian coastline. The objective of this research is twofold: (i) to explore how the definition of homogeneous subsets, based on WP,

affects the estimation of $H_s$ extreme values; (ii) to characterize the identified WPs in the framework of the Mediterranean Region (MR) cyclones climatology.

The paper is structured as follows: in Sect. 2 we introduce the data and describe the methodology developed; results are presented and discussed in Sect. 3; finally, in Sect. 4 conclusions are summarized and further developments are introduced.

## 2 Data & Methods

### 2.1 Wave and atmospheric data

This work takes advantage of eight hindcast points located in the Italian seas, as shown in Fig.1. This choice allowed to test the reliability of the proposed methodology under different local wave climates. In fact, the selected points are differently located along the Italian coastline, and, being exposed to different fetches, they are characterized by peculiar wave conditions. The same locations were taken into account by Sartini et al. (2015), where they performed an overall assessment of the different frequency of occurrence of the extreme waves affecting the Italian coasts. Table 1 reports the names, depths, and coordinates of the selected locations.

The points correspond to as many buoys belonging to the Italian Data Buoy Network (Rete Ondametrica Nazionale or "RON", Bencivenga et al., 2012), which collected directional wave parameters over different periods between 1989 and 2012. Unfortunately, most of the buoys are characterized by significant lacks of data due to malfunctions and maintenances of the devices. We therefore referred to hindcast data, since such a widespread lack of data would imply a loss of reliability for the following analysis. We relied to the hindcast of the Department of Civil, Chemical and Environmental Engineering of the University of Genoa (http://www3.dicca.unige.it/meteocean/hindcast.html; Mentaschi et al., 2013, 2015). It now provides wave parameters on a hourly base from 1979 to 2018 over the whole Mediterranean sea, with a spatial resolution of 0.1° in both longitude and latitude (however, at the time the study was developed the series were defined up to 2016). Data were validated against the records of the buoys (when available); more details can be found in Mentaschi et al. (2013, 2015). The wind data used to drive the wave generation model were derived from the NCEP Climate Forecast System Reanalysis for the period from January 1979 to December 2010 and CFSv2 for the period from January 2011 to December 2018, downscaled over the MR at the same resolution of the hindcast, along with the pressure fields through the model Weather Research and Forecast (WRF-ARW version 3.3.1, see Skamarock, 2009; Cassola et al., 2015, 2016). These wind velocities data were used here to feed the cluster analysis of the wave peaks. It should be pointed out that, in case of sea waves, other variables may concur to affect their propagation (and therefore the bulk parameters), i.e. the local bathymetry and the currents. However, the bottom depth is reasonably expected not to be relevant, since all the locations investigated lie in deep water (see Table 1), while the currents were not fed into the wave model, though the hindcast data were widely validated and proved to be reliable.

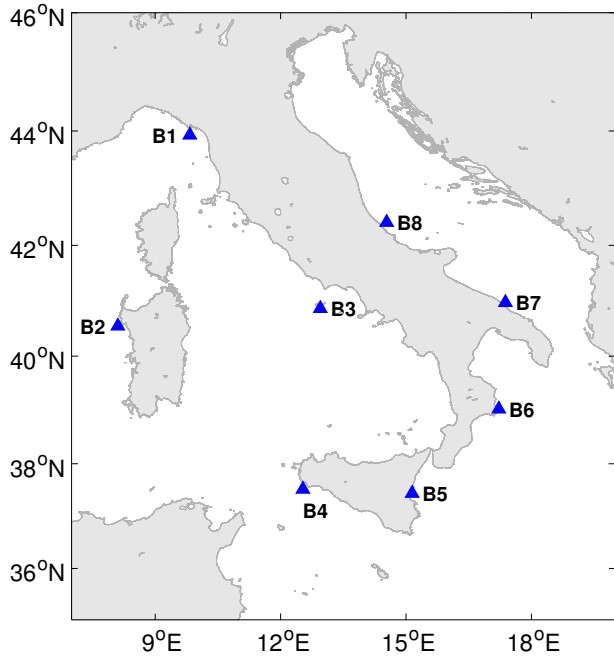

**Figure 1.** Study area and investigated locations with their respective codes

| CODE | LON | LAT | DEPTH [m] | NAME |
|------|------|------|------|------|
| B1 | 9.8278 | 43.9292 | 83.8 | La Spezia |
| B2 | 8.1069 | 40.5486 | 99.7 | Alghero |
| B3 | 12.9500 | 40.8667 | 242.0 | Ponza |
| B4 | 12.5333 | 37.5181 | 90.8 | Mazara del Vallo |
| B5 | 15.1467 | 37.4400 | 65.4 | Catania |
| B6 | 17.2200 | 39.0236 | 611.7 | Crotone |
| B7 | 17.3778 | 40.9750 | 80.0 | Monopoli |
| B8 | 14.5367 | 42.4067 | 55.8 | Ortona |

**Table 1.** Lon/lat coordinates and depths of the hindcast locations employed in the study (reference system: WGS84)

## 2.2 Extreme events selection

For each location, wave height peaks were selected through a Peak Over Threshold (POT) approach, and in particular by using a
90 time moving window. This approach works as follows. First, the whole series of $H_s$ is spanned through a time moving window
of given width; second, when the maximum of the data within the window happens to fall in the middle of the window itself,

it is retained as a peak; finally, in order to get rid of the peaks which are not related to severe sea states, a first $H_s$ threshold is chosen and only peaks exceeding this threshold are retained for further analysis.

In this study, for each location the width of the moving window was set equal to one day, meaning that the inter-arrival time between two successive storms is at least equal to one day. The threshold was fixed as the $95^{th}$ percentile of the resultant peaks. This ensured to maintain a uniform approach for all the locations, efficiently capturing the different features of the local wave climates. Beside the significant wave heights, we retained as well the waves mean incoming directions corresponding to the peaks ($\theta_m$), which were used for analysing the outcomes of the clustering algorithm. Finally, for each peak we extracted the mean sea level pressure field (MSLP) and surface wind fields for several time lags (0, 6, 12 24, 36, and 48 hours earlier with respect to the peak's date), over the whole MR. Wind fields were used to classify the selected peaks due to their parent WP, as described in the following section; MSLP fields were used instead for the post-processing and climatological analysis of the results.

## 2.3   Extreme events classification: definition of weather patterns

The classification of extreme events is based on surface wind fields ($\bar{u}_w$) observed in the whole MR during the hours before and concomitant to the time of the peaks. In order to define the spatial and temporal domains to be taken into account, we looked at the correlation maps between the wind velocities and the $H_s$ peaks for different time lags. Correlations were evaluated over a sub-grid of the atmospheric hindcast, with nodes spaced of $0.5°$ both in longitude and latitude. To compute the correlation between $H_s$ and $\bar{u}_w$ series is not straightforward, as the former variable is scalar and the latter is directional. To tackle this issue, we followed the procedure suggested by Solari and Alonso (2017). Given a time lag $\Delta t$, the wind at every node $(i,j)$ is defined by its zonal and meridional components $(u_x, u_y)_{(i,j,\Delta t)}$; the correlation between $H_s$ peak series and the time lagged surface wind speed at any given node is then estimated as the maximum of the linear correlations obtained by projecting the wind speed series in all the possible directions:

$$\rho_{(i,j,\Delta t)} = \max_{0 \leq \theta < 2\pi} \left\{ \rho\left( H_s ; \bar{u}_{(i,j,\Delta t,\theta)} \right) \right\} \tag{1}$$

where $\rho_{i,j,\Delta t}$ is the resulting correlation for node $(i,j)$ at time lag $\Delta t$, $\rho$ refers to linear correlation function, $u_{(i,j,\Delta t,\theta)}$ is the surface wind speed projected along direction $\theta$ according to Eq (2):

$$\bar{u}_{(i,j,\Delta t,\theta)} = u_{x(i,j,\Delta t)} cos(\theta) + u_{y(i,j,\Delta t)} sin(\theta) \tag{2}$$

in this way not only a maximal correlation is obtained for every node, but also the direction corresponding to the maximal correlation, estimated as:

$$\hat{\theta}_{\rho(i,j,\Delta t)} = \underset{0 \leq \theta < 2\pi}{\mathrm{argmax}} \left\{ \rho\left( H_s ; \bar{u}_{(i,j,\Delta t,\theta)} \right) \right\} \tag{3}$$

The correlation maps computed with Eq. (1), allowed to evaluate the spatial domain and the time lags for which $\bar{u}_w$ is significantly correlated to (i.e., directly affecting) the resulting wave peaks at a given location.

Once the spatial and time domain of the wind fields producing the peak wave conditions were defined, the wind fields were used for clustering and classifying the extreme events. To this end, the *k-means* algorithm was used, fed with the normalized wind fields. *k-means* is aimed at partitioning a N-dimensional population into *k* sets (clusters) on the basis of a sample, in order to minimize the intra-cluster variance (MacQueen et al., 1967). The normalization of the wind fields sought to reduce the influence of the intensity of the wind speed on the classification, so that only the spatial form of the field and its time evolution were taken into account. Note that the values of $H_s$ did not play any role in the classification of the peaks but the identification of the point in time of the wind fields.

## 2.4   Analysis of the WPs climatology

Once the wind fields, and therefore peak $H_s$ series, were grouped into *k* clusters, the MSLP corresponding to the events within each cluster were averaged and the position of the lowest pressure was recorded, for all the $\Delta t$ taken into account. This allowed us to track the paths of the averaged low pressure systems corresponding to each cluster, defining in turn the respective WP.

At a second time, the dynamics of the systems were compared with those of the cyclones typically detected in the Mediterranean sea (Trigo et al., 1999; Lionello et al., 2016), while the frequency of occurrence of the events of different clusters were compared with the outcomes of Sartini et al. (2015). The number of clusters needed to group the series into was defined for every location by looking at the outcome of the cluster analysis: when an increase from $k$ to $k+1$ clusters did not further lead to a new clearly differentiated WP, the research was stopped and $k$ was used for the cluster analysis of that particular location.

## 2.5   Extreme value analysis

The EVA were performed independently over the subsets of $H_s$ peaks resulting from the cluster classification. We followed the methodology proposed by Solari et al. (2017), where the threshold for the POT analysis is estimated as the one maximizing the $p_{value}$ of the upper tail Anderson-Darling test. Then, once the threshold is identified (i.e. a subset of the peaks within a given WP is defined), the three parameters of the GPD are estimated through the L-moments method, and the return-period ($T_r$) quantiles of the variable under investigation are computed. In this work, in order to estimate the overall $T_r$-$H_s$ curve and its confidence intervals from the GPDs fitted to each WP, a bootstrapping approach was implemented ( Table 2). The algorithm in Table 2 shows a pseudocode summarizing the bootstrapping procedure. First, $N_{boot}$ series of $H_s$, each $N_{years}$ long, are generated for every WP. Second, the series generated for the different WPs ($N_{WP}$ per location) are combined in order to obtain one single $N_{years}$ long series for each of the $N_{boot}$ simulations. Third, an empirical relation between $T_r$ and $H_s$ (i.e. an empirical cumulative distribution function or ECDF) is estimated from each one of the $N_{boot}$ series. Lastly, expected value and confidence intervals of $H_s$ are estimated from the $N_{boot}$ ECDFs for several return periods.

The method assumes a Poisson-GPD model for each WP and that the realizations of different WP are independent from each other. This independence hypothesis was evaluated by estimating the correlation between the annual number of peaks associated to each WP.

**Algorithm:**

**for** $j$ from 1 to $N_{boot}$ **do**

    **for** $i$ from 1 to $N_{WP}$ **do**

        Randomly simulate $\{H_s\}_{i,0}$ (a vector of $N_i$ values of $H_s$) from distribution $GPD(\hat{\theta}_{(i,0)})$

        Estimate $GPD$ parameters $\hat{\theta}_i$ from $\{H_s\}_{i,0}$

        Randomly simulate $\{N_{simu}\}_i$ (number of events in a $N_{years}$ length simulation) from a Poisson distribution with parameter $\lambda_i$

        Randomly simulate $\{H_s\}_i$ (a vector of $\{N_{simu}\}_i$ values of $H_s$) from distribution $GPD(\hat{\theta}_i)$

    **end for**

    Combine all vectors $\{H_s\}_i$ $(i = 1, \ldots, N_{WP})$ into a single vector $\{H_s\}_j$

    Estimate empirical $H_s - Tr$ curve from $\{H_s\}_j$

**end for**

Estimate confidence intervals from the $N_{boot}$ empirical $H_s - Tr$ curves

**Parameters:**

$N_{boot}$ is the number of bootstrapping repetitions

$N_{WP}$ is the number of WP

$\hat{\theta}_{(i,0)}$ are the parameters of the GPD estimated from the original sample

$N_i$ is the length of the original sample of $H_s$ peaks within the WP

$\hat{\lambda_i}$ is the yearly number of events of the i$^{th}$ WP

$N_{years}$ is the number of years simulated; it must be larger that the maximum return period to be analyzed

$N_{simu}$ is the number of events in $N_{years}$ obtained for the i$^{th}$ WP

**Table 2.** Computational scheme of empirical extreme values curves and its confidence intervals

For a given location, the overall work-flow can be summarized as follows:

- selection of a series of $H_s$ peaks through a POT approach

– selection of the wind field data to be employed in the clustering algorithm (i.e. $\Delta t$ and spatial domain)

- classification of the $H_s$ peaks due to the $k - means$ algorithm

- definition of a suitable number ($k$) of WPs

- averaging of the MSLP corresponding to the peaks of each WP, for each $\Delta t$ taken into account

- performing EVA over the single subsets

– computation of the overall long term distribution through a bootstrapping technique

This methodology was applied to all the hindcast locations shown in Fig. 1. In this paper, for the sake of clarity, just the results of the locations B4 (Mazara del Vallo) and B7 (Monopoli) are shown and discussed. Indeed, just one WP was necessary

for classifying the peaks in La Spezia and Alghero (B1 and B2), thus no further analysis were performed. Among the locations left, we simply selected the locations furthest from each other in the East-West direction. However, the results related to the other locations can be found in the supplementary material.

## 3   Results and Discussion

Once the series of $H_s$ peaks is selected, the first step of the proposed methodology requires to define the domains of $\bar{u}_w$ in time and space due to the outcomes of the correlation analysis between the two parameters. Indeed, as mentioned in Chapter 2.1, the wind is reasonably expected to play the mayor role for the observed wave parameters at the investigated locations. Figure 2 shows the correlation maps for B7 for different time lags, along with the directions leading to the maximum values of correlation in each node. It is interesting to see how $\hat{\theta}$ for $\Delta t$=0 hours are distributed along the nodes characterized by similar values of $\rho$. Actually, even though the values of $\hat{\theta}$ come from a purely statistical analysis (i.e., they were computed with Eq. (3)), their spatial distribution follows that of a typical cyclone. Velocities happen to be uniformly oriented along the nodes characterized by the higher values of $\rho$, close-to-tangential to a circle centred on the nodes showing instead lower values of $\rho$. This allows to get a first insight on the predominant process most likely affecting the wave climates of the investigated locations, as it will be discussed further on this paper. On the contrary, the analysis of the correlations between $H_s$ and $\bar{u}_w$ reveals that the areas characterized by similar values of $\rho$ are not uniformly distributed in the neighbourhood of the points considered. It is therefore difficult to uniquely contour the nodes to be taken into account for the successive analysis. As regards the time step, correlations rapidly decrease for $\Delta t$ longer than 12 hours, after which no evidence of significant $\rho$ can be observed. The same outcomes apply for all the other locations taken into account. Results suggest that the events shall be linked with broader circulation patterns. In view of the above, it was decided to refer to the whole MR and time lags of 0 and 12 hours for the purposes of peaks classification.

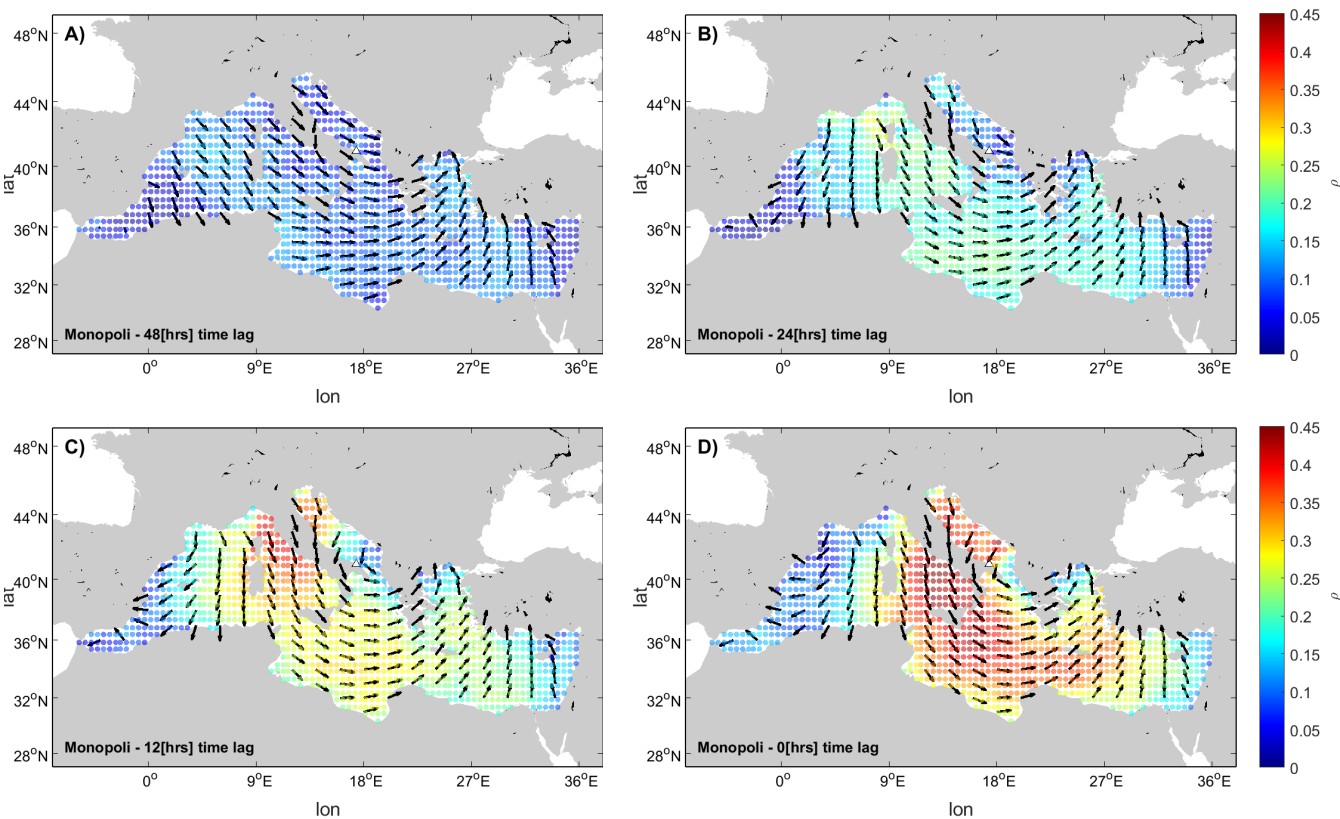

**Figure 2.** Correlations between $H_s$ and $\bar{u}_w$ in location B4 for different time lags. Panel A): $\Delta t$ equals 48 hours; panel B): $\Delta t$ equals 24 hours; panel C): $\Delta t$ equals 12 hours; panel D): $\Delta t$ equals 0 hours

Once the spatial and temporal limits of the wind fields to be used in the $k-means$ were defined, we performed a sensitivity analysis over the resultant average MSLP belonging to each cluster, among the total amount of tested clusters ($k$). Two clusters were needed to detect different systems at all the sites but Alghero and La Spezia, where the local extreme waves could be related to a single pattern. For the other locations, to increase $k$ did not lead to systems significantly diverging from those already defined. The reason for such a small number of resulting WPs has to be found in the nature of the $H_s$ employed in the analysis. Indeed, these are associated to extreme sea states, which are most likely driven by atmospheric phenomena developing along well-defined and fixed tracks.

Figures 3 and 4 show the averaged pressure fields corresponding to both the clusters and $\Delta t$ in B4 and B7 hindcast points. Here, two main systems can be clearly distinguished: i) a low moving SW-NE towards the Balkan area (say WP#1) and ii) a low moving NW-SE (henceforth referred to as WP#2).

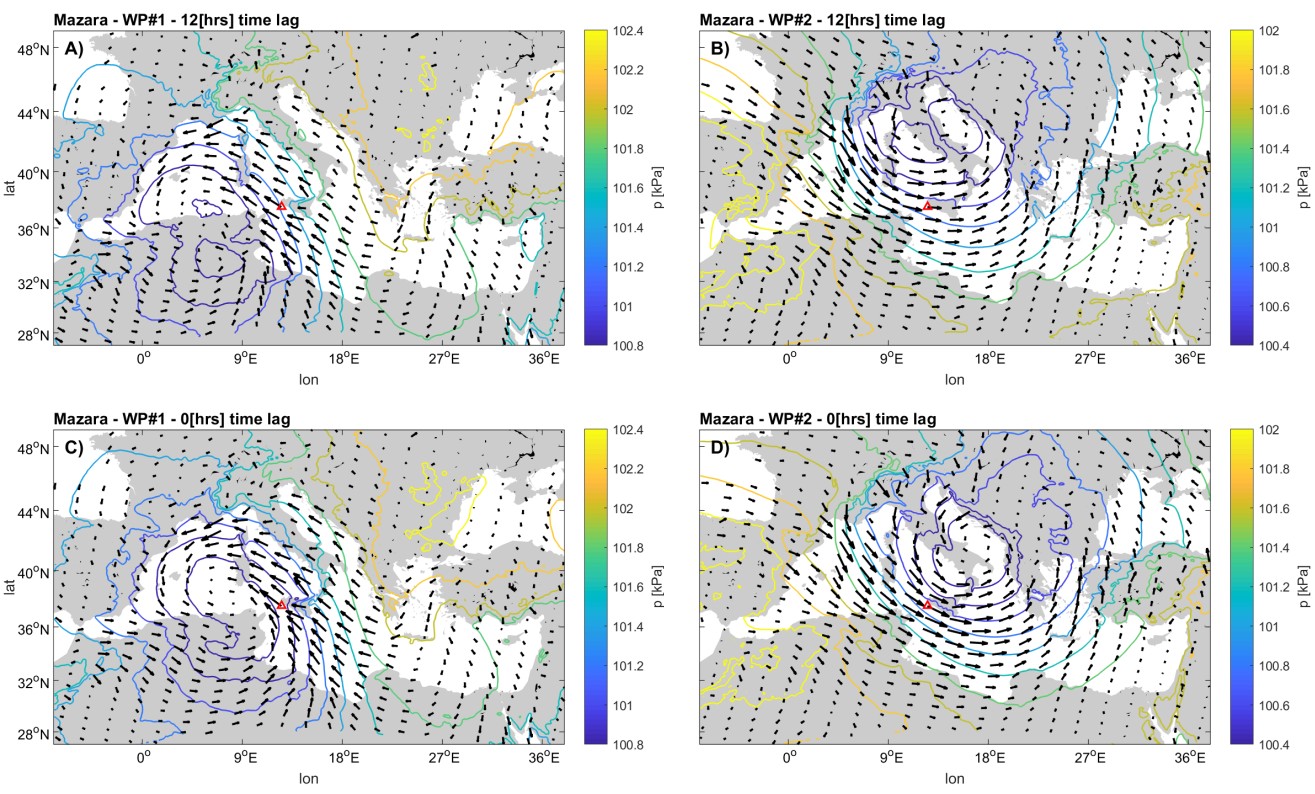

**Figure 3.** Average MSLP for the $H_s$ peaks in Mazara del Vallo (B4). Panel A): WP#1, $\Delta t$ equals 12 hours; panel B) WP#2, $\Delta t$ equals 12 hours; panel C): WP#1, $\Delta t$ equals 0 hours; panel D): WP#2, $\Delta t$ equals 0 hours

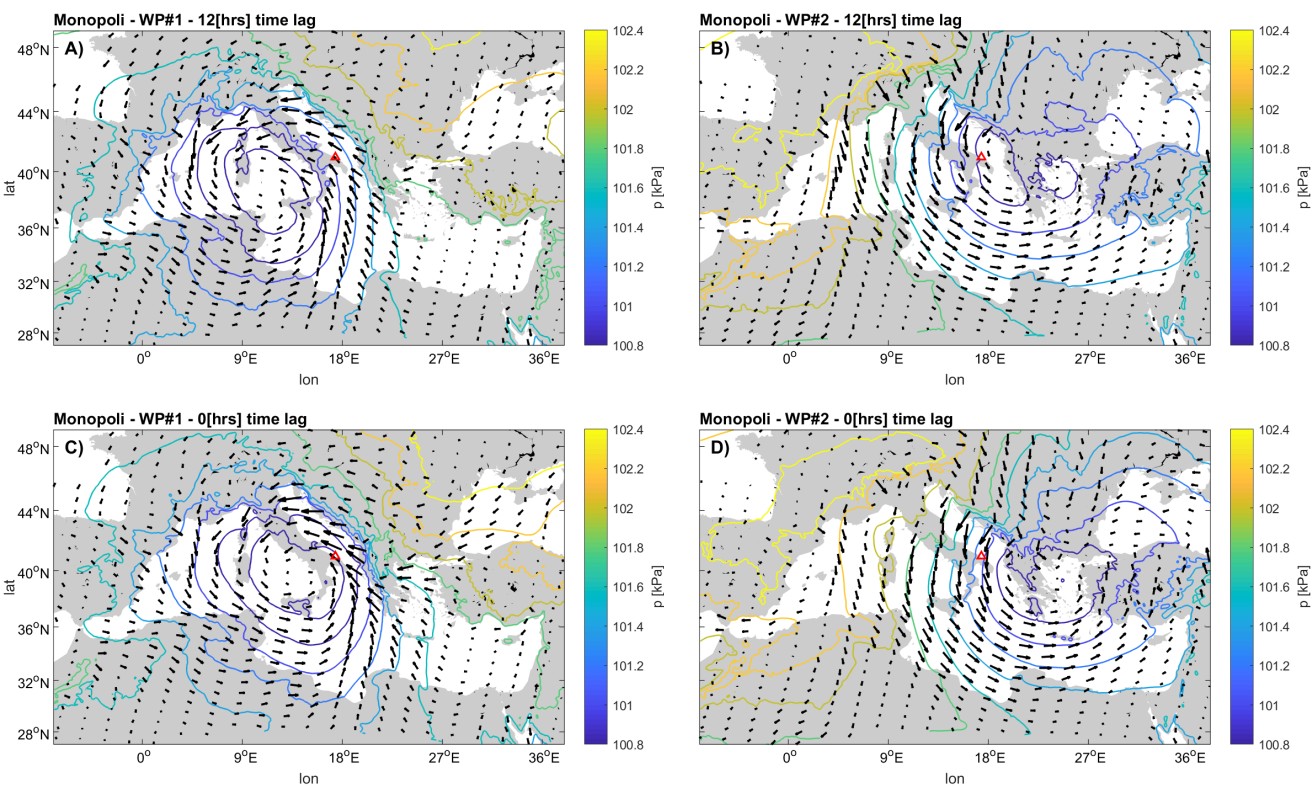

**Figure 4.** Average MSLP for the $H_s$ peaks in Monopoli (B7). Panel A): WP#1, $\Delta t$ equals 12 hours; panel B) WP#2, $\Delta t$ equals 12 hours; panel C): WP#1, $\Delta t$ equals 0 hours; panel D): WP#2, $\Delta t$ equals 0 hours

Let us first focus on WP#2. Low pressure moves SE from the central Europe, crossing the Adriatic sea and decreasing its intensity once it gets to the south Balkan area, where it stops until it finally dissolves. As regards WP#1, the cyclogenesis most likely takes place in the east area of north Africa, with cyclones first approaching the west coastline of Italy while moving NE. The paths of WP#1 and WP#2 show interesting similarities with well known cyclones typically forming and departing from two of the most active cyclogenetic regions in the MR, respectively the lee of the Atlas mountains and the lee of the Alps (Trigo et al., 1999). The MSLP composites related to WP#1 and WP#2 are consistent with those highlighted in previous research, for instance Lionello et al. (2006), showing the synoptic patterns associated with extreme significant wave heights in different regions of the Mediterranean Sea (see in particular the MSLP fields reported in Fig. 122, panel A) and D) for WP#2 and WP#1, respectively). In particular WP#2 seems to follow the Genoa system path, that usually moves down to the Albanian and Greek coasts, while that of WP#1 shows characteristics analogous to the Sharav depression, moving north-eastward toward the Greek region (Flocas and Giles, 1991; Trigo et al., 1999, 2002).

At a second time, the MSLP related to the three highest events for both the locations and the WPs were individually analyzed, 205 in order to evaluate their homogeneity with respect to the overall MSLP averages. Results are shown in Fig. 5 and Fig. 6.

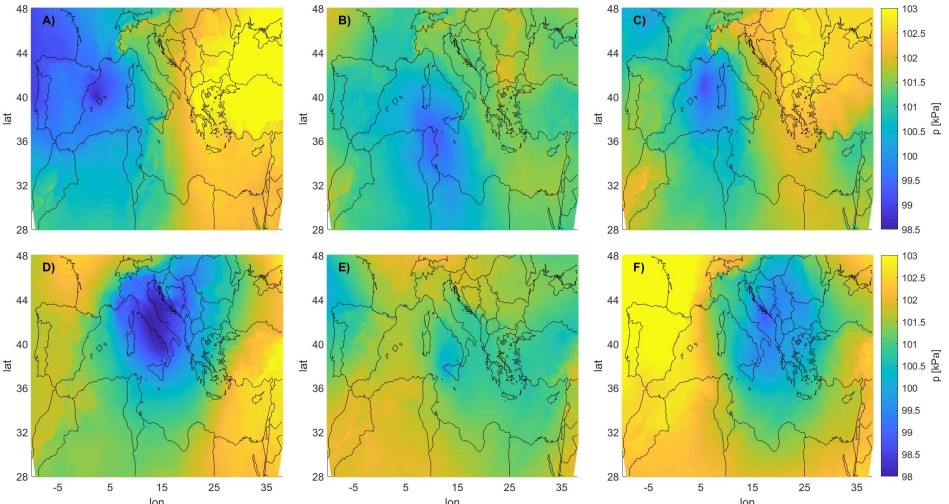

**Figure 5.** MSLP associated to the three highest POT events in Mazara del Vallo (B4). Panels A), B), and C): WP#1 events; panels D), E), and F): WP#2 events. The storms are sorted from left to right in decreasing order

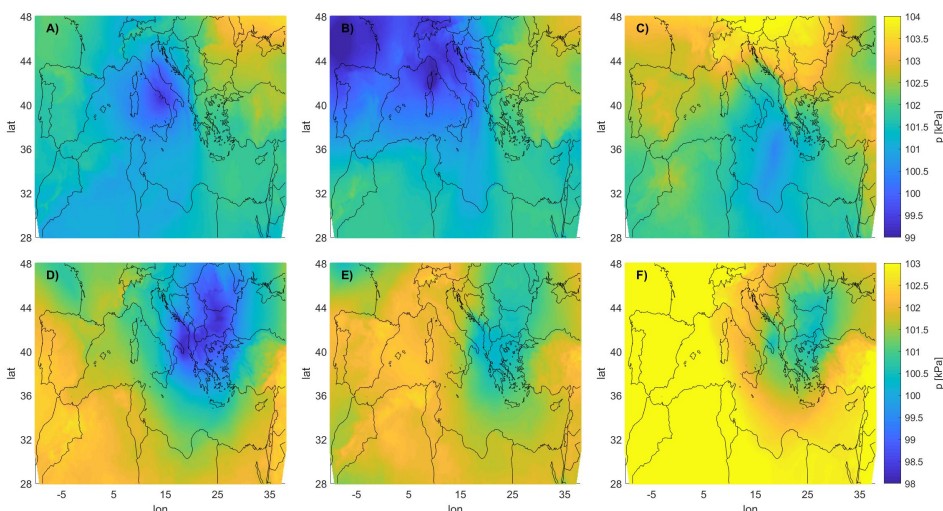

**Figure 6.** MSLP associated to the three highest POT events in Monopoli (B7). Panels A), B), and C): WP#1 events; panels D), E), and F): WP#2 events. The storms are sorted from left to right in decreasing order

From the MSLP charts of Fig. 5 and Fig. 6, it can be noticed how low pressures are in good agreement in terms of their locations, and in turn to the average MSLP fields reported in Figures 3 and 4. Actually, this especially applies in case of the WP#2 events: the location of the low pressure of the cyclone is similar for the analyzed storms, though there are differences in terms of the absolute value of the pressure and slightly in the shape of the cyclone too (the values of the color scale were

210 modified to appreciate better the locations of the lows). This is something to be expected, as it partially explains the different intensities of $H_s$ (of course, local effects have to be taken into account as well). However, there is an important degree of variability within each family of events, in particular as regards WP#1.

The paths highlighted for WP#1 and WP#2 characterize the lows of the WPs detected in all the investigated sites (see the supplementary material). A summary of the lows position at the two different $\Delta t$ can be appreciated in Fig 7, reporting the

215 tracks of both the WP#2 and WP#1 systems in all the sites but B1 and B2; indeed, in the latter cases the analysis of the MSLP fields did not allow to identify two separated systems.

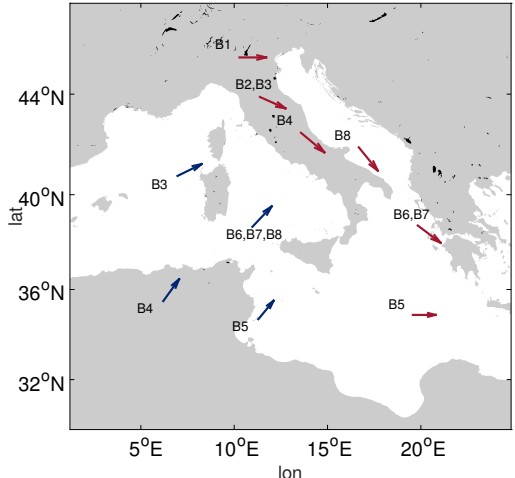

**Figure 7.** Time evolution of the center of the low pressure for the different WP identified for each buoy. In blue WPs with lows traveling northeastward (WP#1); in red WPs with lows traveling east-or southeastward (WP#2)

It is interesting to see how the WP#2 low get across the investigated sites in a precise chronological order, crossing first the northernmost locations and then those next to the south Balkan area where the cyclone actually ends its run. As such, we took as a reference four buoys affected by WP#2 at different times, evaluating the time lag between their respective peaks generated by such system. We considered the extreme series of B4, computing for each storm the time lag between the peak at the buoy

and peaks occurring in B2 and B1 (occurring earlier) and B6 (occurring later); distributions of the time lags are shown in Fig. 8.

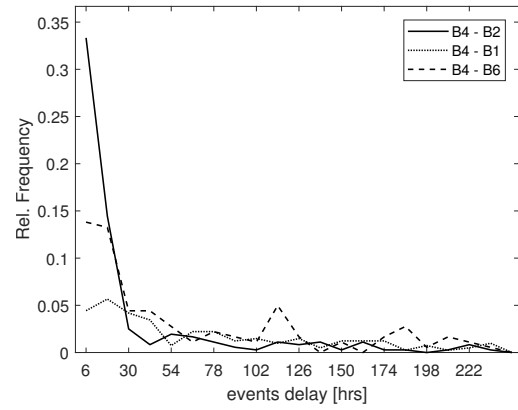

**Figure 8.** Relative frequency of the events' delay for B2, B1 and B6. Referring series is that of B4

Looking at Fig. 8, it is evident how the majority of the events' delays between B4 and the other investigated points fall within about forty hours. We therefore evaluated the average MSLP fields for time lags up to 48 hours before and after the storms occurring at the reference location, tracking the low pressure centre evolution. Results show how the cyclone runs out in a couple of days, the center of the low travelling for approximately 1600 kilometres, with a resulting speed of ∼33 km h$^{-1}$ (see Fig. 9). Both the lifetime of the identified cyclone and the speed it moves at are compatible with the features of the cyclones most frequently encountered in the MR (Lionello et al., 2016). Unfortunately, as regards WP#1, an equally clear path cannot be detected, since the average lows apparently arise simultaneously in most of the points taken into account. As previously pointed out, the MSLPs related to this pattern show a higher variability with respect to those characterizing the events of WP#2, thus they would require further deepening and more detailed investigations. Therefore, we did not characterize the mean evolution of the MSLP fields related to the events of WP#1.

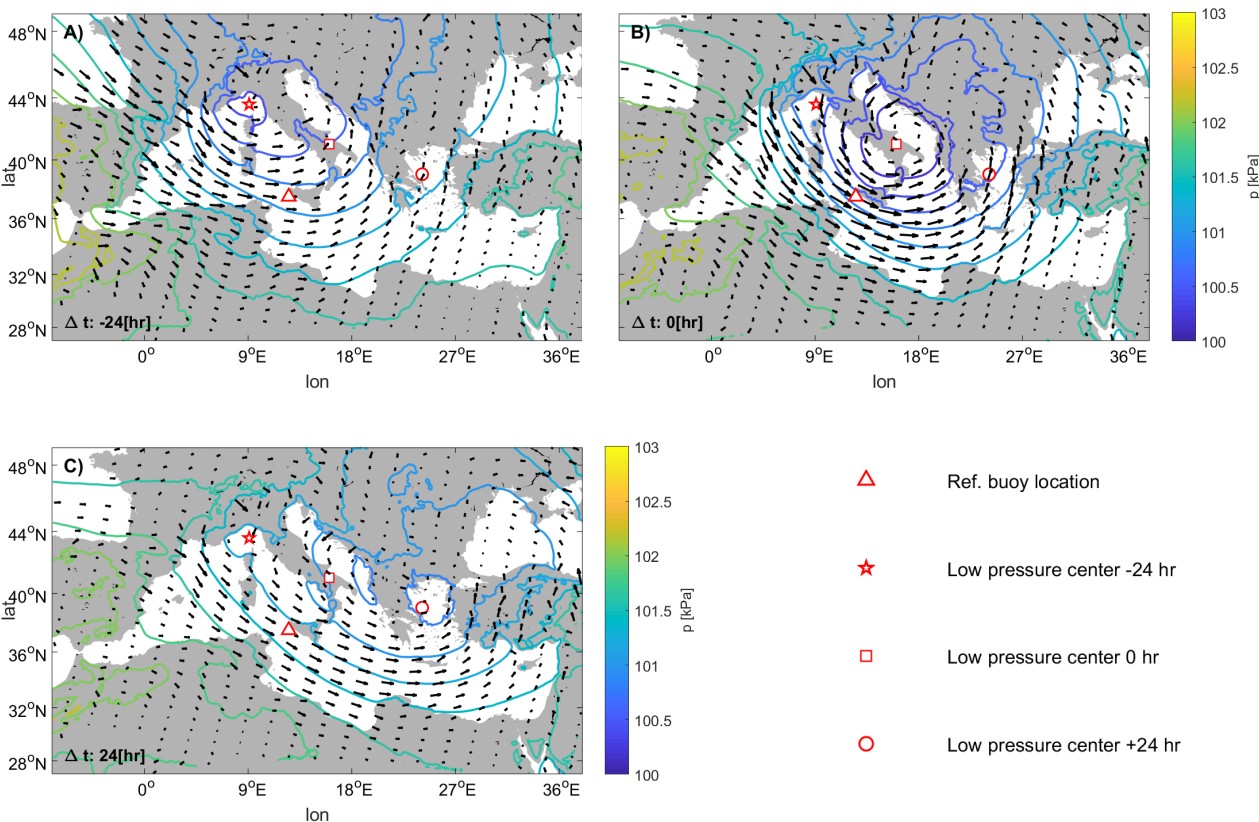

**Figure 9.** WP#2: average MSLP evolution with respect to the reference dates of the events in B4 (underlined with the red triangle)

The characterization of the two systems is reflected in the frequency of occurrence of the storms, with peaks belonging to different subsets showing as well distinctive seasonality. From the results shown in Fig. 10, it can be noticed how WP#2 peaks mainly occur in winter, whereas the events of WP#1 are characterized by two milder intra-annual peaks of occurrence, spread among the spring and autumn months. The intra-annual cycle of the WP#1 events further suggest a direct link with the Sharav cyclones, which show similar seasonal fluctuations; as regards the WP#2 peaks, even though the storms of the Genoa low are more uniformly distributed along the year, the most intense events precisely occur during winter, as it happens in the above mentioned locations (details can be found in Lionello et al., 2016).

Figure 11 summarizes the results of the monthly frequency of occurrence of the extremes in all the investigated locations, grouped according the WP. For each location, frequencies were normalized in the 0-1 space with respect to the total amount of peaks, in order to be able to compare outcomes defined over different ranges.

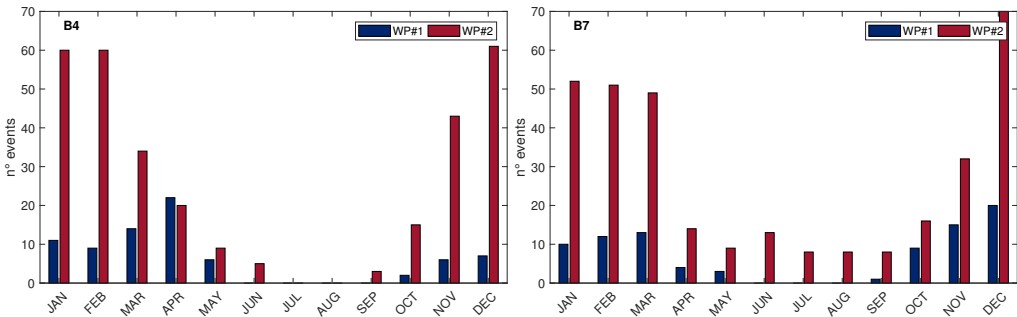

**Figure 10.** Monthly number of events for different WP. Left panel: B4; right panel: B7

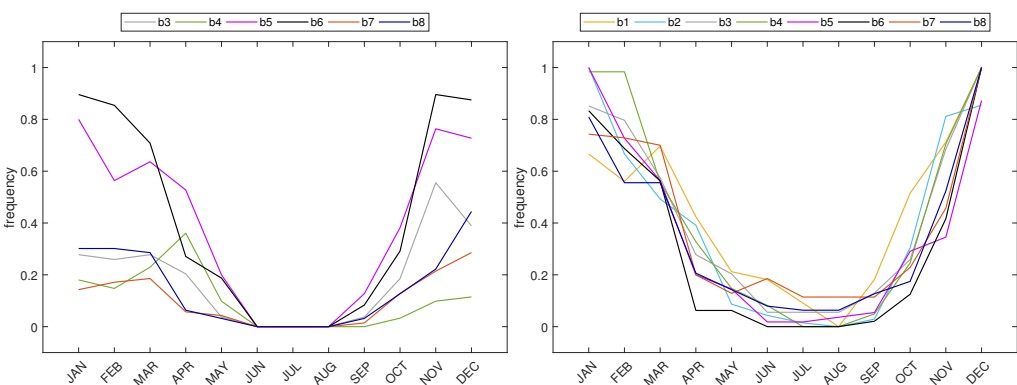

**Figure 11.** Normalized monthly frequency of occurrence of the extremes. Left panel: events belonging to WP#1; right panel: events belonging to WP#2.

It can be noticed how the relative weight of WP#1 events on the overall peaks distributions increases moving south. Points in the northernmost locations (B1 and B2) apparently are not influenced by the WP#1 system, and their extreme waves are actually linkable just to WP#2. Moving to the southernmost locations (B5 and B6), no significant differences in the seasonality of the two patterns' events can be appreciated; in these two buoys none of the two systems shows a prevailing frequency of occurrence for the induced storms.

The aforementioned behavior may be justified by looking at the location of the investigated points (see Fig. 1): lows moving north-eastward directly run over B5 and B6, marginally affect B7, B8, B4 and B3, whereas they do not interest B1 and B2. However, position of the buoys with respect to the cyclones' paths is not the only relevant variable. Indeed, local bathymetry and prevailing fetch characteristics may result in storms having, on average, unique characteristics, e.g. it may happen that

peaks related to the same WP show distinct frequencies of occurrence, for instance the events related to WP#1 in B6 and B7. In this latter case, the predominant parameter seems to be the fetch length, which for B7 is very limited with regards to the NE incoming waves (precisely related to the first weather circulation pattern).

The WPs that are defined in the present study show common characteristics with those qualitatively identified by Sartini et al. (2015) in a seasonal variability analysis of extreme sea waves. In particular, the same WP was identified in B1 and B2, linking the extremes with the Gulf of Genoa cyclogenesis WP type. As Sartini et al. (2015) noted, even if cyclones in the Genoa Gulf are a constant feature over the whole year, those connected at the extreme events are the ones characterized by the lowest value in MSLP. These findings (i.e., higher values during the winter rather than in spring and summer time) were observed in
the southern Thyrrenian Sea as well (for instance in B4) and in the Central Thyrrenian Sea (B3). In the latter case, we found a more marked seasonal variation of the extremes, as the two resultant WPs are well separated between winter and autumn. Intra-annual variability was observed also for B7, while the analysis carried out by Sartini et al. (2015) did not reveal this kind of behaviour; analogously, B5 and B8 buoy revealed the presence of two distinct WPs, while the previous analysis identified just one cyclogenesis system. These differences may be justified in the first place by the different peaks selection (a moving
window for this study, while Sartini et al., 2015, use a partial duration series approach). Moreover, evaluation of the seasonality follows completely different algorithms: the present study directly applies a clustering technique over the selected peaks, while the former analysis used the peaks to model a time dependent distribution, further characterizing their seasonality on the basis of the values of the distribution coefficient aroused from the fitting procedure.

Another interesting outcome regards the characteristics of the extremes differentiated due to the parent clusters following
the $k-means$ algorithm. Figure 12 shows the covariates $H_s$ and $\theta_m$ of the selected peaks belonging to each WP.

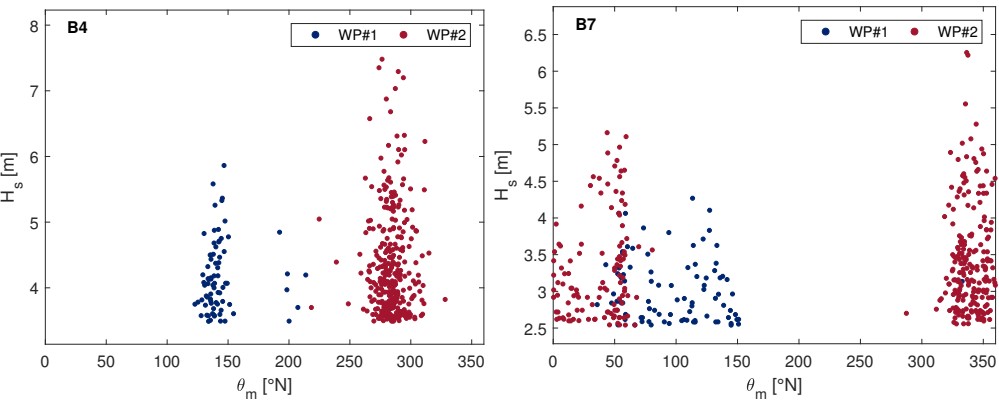

**Figure 12.** Scatter plot of $H_s$ and $\theta_m$ due to different WP. Left panel: B4; right panel: B7

Looking at the scatter plots, it can be appreciated how the events belonging to different clusters are remarkably homogeneous in terms of wave bulk parameters, even though the latter were not used for clustering the peaks. In case of B4, where is clear a bimodal distribution with respect to the waves' incident direction (i.e. two peaks corresponding to SE and NW), the proposed

methodology allows to differentiate the sea storms according to the directional frames of the local wave climate. On the other hand, when the waves' direction is more uniformly scattered over a given sector, it is not unusual that different climate forcing result in the same wave direction; this can be noticed in B7, where $\theta_m$-$H_s$ scatters belonging to different patterns are partially overlapped, thus a directional classification is not straightforward and most probable would not be able to differentiate the storms due the cyclones generating them. Nevertheless, also in such a case the distributions of $H_s$ are homogeneous within each WP, with the more severe $H_s$ related to WP#2 system in both the locations.

Finally, Fig. 13 shows the $T_r$-$H_s$ curves, comparing the results obtained directly from the whole set of peaks with those obtained from the single-WP distributions, combined by means of algorithm given in Table 2 (i.e., omni-WP curves).

The omni-WP curves show a remarkable agreement with those carried out through the analysis of the whole dataset without WP classification; in both the locations B4 and B7 it can be even appreciated a narrowing of the confidence intervals, meaning a reduction in the total variance for the long-term estimates. Actually, the curve related to B4 show a slight deviation between the two approaches ($\simeq$ 30 cm over the 200 year wave); however, such small magnitudes imply relative errors of $\simeq$ 4% with respect to the omni-WP curves, and can be therefore considered as an uncertainty inherent in this kind of computations (see e.g., Borgman and Resio, 1977, 1982, stating that reliable long term estimates can be carried out just up to three times the years the length of the original dataset).

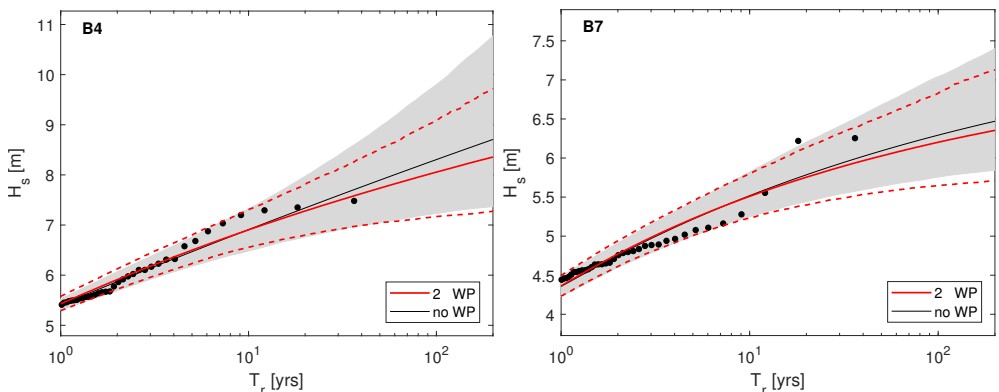

**Figure 13.** Omni-WP extreme value distributions of $H_s$ obtained from the whole set of peaks (black) and from combining single-WP distributions (red), along with 90% confidence intervals (grey shadow and red dashed lines, respectively). Left panel: B4; right panel: B7

These results seem to reinforce the validity of developing the omni-WP curve under the hypothesis of independence between the events of different WPs. In this case, the independence hypothesis was to some extent corroborated by the low correlations values attained by the intra-clusters annual frequency of occurrence for all the locations: -0.17, 0.23, -0.13, -0.11, -0.34, 0.13 for B5, B6, B4, B7, B8 and B3 respectively). In conclsion, it is good to recall that thresholds for the EVA were selected in order for the peaks to come from a Generalized Pareto family (Solari and Alonso, 2017), thus guaranteeing the intra-cluster homogeneity of the data.

## 4 Conclusions

In this work, the extreme sea storms at eight hindcast points differently located along the Italian coasts were analyzed. The investigated locations are characterized by different conditions, both in terms of their local orography and exposure, and this is consequently reflecting on the respective wave climate showing, on average, peculiar characteristics. However, despite the differences arising due to the local effects, the analysis here introduced reveal how the most severe storms among all the locations can be related to particular atmospheric circulation patterns, defined at the synoptic scale. These patterns result in homogeneous features of the wave peaks at the investigated locations, both in terms of frequency of occurrence and significant wave heights. Such features suggest that the extreme distributions of $H_s$ can be singularly evaluated for each WP, the starting datasets being homogeneous and independent with respect to each other.

The methodology here introduced allows the classification of extreme wave events (or other oceanic variables) into homogeneous subgroups according to the circulation patterns most likely generating them. Starting from local extreme waves, the analysis is extended to a basin scale, resulting in a reduced number of circulation or weather patterns.

Such an approach might facilitate the physical interpretation of sea storms, as well as their linkage with the climatology of the basin. In particular, for the analyzed locations, the proposed methodology led to the identification of two different cyclonic systems, characteristic of the atmospheric circulation on the Mediterranean Sea, as the possible origin of the extreme wave events affecting the Italian shores. Two well-known circulation patterns were highlighted: one characterized by a low departing from the mid western Europe and moving south-east (referred to as WP#2); the other forming in the lee of the Atlas and crossing the Mediterranean sea north-easterly (referred to as WP#1).

When extreme events are classified according to their meteorological origin, there is a great confidence of working with homogeneous samples, thus being in compliance with the main hypothesis underlying the EVA (Mathiesen et al., 1994; Coles, 2001). As such, return levels can be computed independently for the events belonging to each pattern identified, and the overall long-term distribution of $H_s$ can be computed starting from the single distributions fitted to each subset with no loss of information. The method proposed relies on a Monte-Carlo simulation, and it is shown how, in our study case, the divergences arising between the outcomes of usual EVA scheme and those following the initial classification of the peaks are negligible. Indeed, in general the distributions we obtained by following the two different approaches was very similar, and in some cases a narrowing in the confidence intervals due to the initial clustering of the data was even achieved. However, it is not straightforward to generalize this conclusion to every possible location, as the difference between analyzing separately the different WPs populations and analyzing all the data pooled together would depends on the characteristics of the different populations at the site.

The method, as presented here, does not contemplate the inclusion of trends and inter- or intra-annual cycles. However, the extension of the methodology in this direction is straightforward, as the methods previously developed for non-stationary analysis (see e.g., Sartini et al., 2015) could be applied without major complexities to each of the subsets that are obtained from the classification. On the contrary, it is not obvious how to proceed in the selection of the number of patterns to be considered. Here this number was chosen following a qualitative analysis of the results, which was viable for the case study analysed,

though this approach is not always feasible. Indeed, sometimes it might be necessary to (at least) resort to a sensitivity analysis
of the results, as long as a quantitative methodology is not available for the definition of the number of clusters.

Finally, it is worth to mention that the classification of the peaks could facilitate other aspects of the analysis not included in this work, as for instance the multivariate analysis of extreme events. In such a framework, to classify the wave fields according to the wind velocities leads to clusters of $T_p$ consistent with those of $H_s$, as the latter parameter is closely tied to the former one (especially in case of extreme sea states).

*Code and data availability.* The algorithms used in this work were developed in Matlab®. The codes and the hindcast data employed are available upon request. Please contact FDL at: francesco.deleo@edu.unige.it.

*Author contributions.* FDL developed the algorithms and wrote the paper. SS coordinated the work and contributed to developing the codes and writing the paper. GB provided the data, analysed the results in the frame of the MR climatology, and revised the writing of the paper.

*Competing interests.* The authors declare that they have no conflict of interest.

*Acknowledgements.* Mobility of Francesco De Leo has been financed by the University of Genoa through the program "Fondo Giovani 17/18".

Authors deeply acknowledge Francesco Ferrari of the University of Genoa for the help in processing the meteorological hindcast.

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
