# Peer review of "Extreme waves analysis based on atmospheric patterns classification: an application along the Italian coast"

_Natural Hazards and Earth System Sciences, 2019_

## Referee Comment (RC1) · Anonymous Referee #1 · 16 Nov 2019

In the manuscript "Extreme waves analysis based on atmospheric patterns classification: an application along the The Italian coast" the authors propose a methodology for classifying data of a physical quantity, to be applied prior to perform Extreme Value Analysis, for complying with three key requirements of the data samples: independent and indentically distributed and, in addition for directional variables, grouped in homogeneous subsets. This last requirement is the principal objective of the manuscript, applied to significant wave height peaks along the Italian coast. Following previous works, they propose to use the atmospheric processes producing extreme wave conditions in a given location, (1) to select the homogeneous subset based on the weather patterns (WPs), and (2) to estimate the overall extreme values distribution starting from

the distribution fitted to each subset.

The method rely on the physical connection between the atmospheric processes, spatial and temporal evolution of the surface pressure and wind fields, behind the ocurrence of the extreme wave conditions at a given location. Consequently, the classification of extreme events is based on observed surface wind fields during the hours before and concomitant to the time of the peaks and on the correlation maps between the wind velocities and significante wave height peaks. Once the wind fields producing the peak wave conditions are identified the wind fields were used for clustering and classifying the extreme events. Then, the classification of the peaks, once the threshold of the wave height is chosen, depends only on the normalized wind fields.

The manuscript addresses relevant scientific and technical questions within the scope of NHESS, wave climate extreme value problem, presents new data and some novel concepts, with well developed tools and very interesting results. Then it is worthy to be published in NHESS. However, before to recommend the manuscript for publication, there are, in my opinion, two important questions which the authors should clarify:

(a) The range of validity of the classification of extreme events, only based on surface wind fields. (b) The "quality" of the obtained homgeneous data sets resulting from the feeding the k-means algorithm with the normalized wind fields producing the wave height peak conditions.

Developed Questions The main difference between previous published research and present work is that the variable wind wave is generated under fecth and time limited conditions. Then, the correlation between observed wind velocity and wave height fields depends on the generation process quantified through the non-dimensional variables. Significant wave height and peak period, $Hs^*, Tp^*$ depend on the non-dimensional fetch, $F^*$ and the non-dimensional time, $t^*$

$Hs^* = gHs/U^2 = f(gF/U^2, gt/U)$ $Tp^* = gTp/U^2 = f(gF/U^2, gt/U)$ $F^* = gF/U^2$ $t^* = gt/U$

U is the mean wind velocity, at a certain height, over the fetch F. Fetch, and consequently U are defined based on the mean wind direction (well identified by the authors, figure 10).

These functional relationships between the non-dimensional quantities should be used to link the weather patterns, wind velocity and and significant wave height time series. Based on that, F* and t* should be relevant quantities (bring in the physics of the wave generation process) to classify the extreme events and the correlation maps for different lags. For that purpose the procedure defined by eqs (1) –(3) should be applied, not with wind velocity but with the non-dimensional quantities, and as well as to feed the k-means algorithm for finding the homogeneous subset.

In addition, but very important issue, the non-dimensional quantities for each direction will help to define asymptotic values of the extreme distributions, as seems to occur figure 11. In location B4, it seems that there is an upper limit around Hs=8 m. This trendcan be checked working with the non-dimensional quantities for bounded values of F and t for that direction. Similarly for location B7, the two hifhest data points (Hs > 6m) depart approximately a 15% from the third largest value of Hs. Belong to the same subset of data? Please use the non-dimensional quantities for checking the homogeneity of the subsets.

Finally, while working with the wave height and neglecting the wave period, any risk analysis on the coastline or relevant maritime structures would be not complete. By using non-dimensional quantities the values Hs and Tp are computed silmutaneously because they depend on F* and t*. Please, if possible include the peak period to complete the necessary information for developing a risk analysis.
* * *

---

## Referee Comment (RC2) · Anonymous Referee #2 · 10 Dec 2019

The study described in the manuscript "Extreme waves analysis based on atmospheric patterns classification: an application along the The Italian coast" proposes a revision of previously developed works focused on exploiting Weather Patterns (WP) classification methodologies for defining homogeneous dataset to be further employed in Extreme Value Analysis (EVA). First of all I would like to acknowledge the effort made by the authors, together with a clear, concise and well-structured presentation of their work. The manuscript addresses scientific issues within the scope of NHESS, such as the extreme value analysis, together with the development of interesting concepts and tools. Nonetheless the approach sounds too simplistic throughout the entire presentation. Although at large scale the general atmospheric circulation pattern can

be described via the proposed technique, the main concern is how the application of such a methodology can be suitable to address extreme waves analysis. In fact, not all the extreme events can be associated to a clear weather pattern considering that the effectiveness of a cyclone to produce an intense marine storm in a basin like the Mediterranean Sea, and along the Italian coast in particular, crucially depends on its position, whose effects are strongly dependent on the interactions with the local orography. Despite not being an expert in the field of WP recognition, I must point out that the proposed methodology for the classification of extreme events in the Mediterranean Sea may not be the proper approach to capture the characteristic wave climate, because of the, possibly high, local scale of the physical processes involved, especially when refering to extreme events, that prevents from identifying the necessary details. For instance, the strong effects caused by the storm that hit the Gulf of Genoa on October 2018 could have been hardly captured by this methodology, though possibly classified as one of the most severe events over that area. Moreover the assumption of making the peak classification depending only on the corresponding wind fields seems to totally underestimate the contribution of other relevant components in determining the characteristic wave climate. In addition, within the Data and Methods section, the spatial resolution of the hindcast employed as reference for the study is not mentioned. The same for the wind input derived from the NCEP Cimate Forecast System Reanalysis. Although the references have been cited, this information would be relevant also to appreciate the overall quality and confidence limits of the anaysis that have been performed. For these reasons, I do not suggest that the paper is published in the present form, suggesting to reconsider the overall approach and make a thorough revsision of the applied methodology, together with the definition of the related confidence limits.

---

## Author Comment (AC1) · 20 Jan 2020

Dear Editor,
we would like to thank the reviewer for further reading and commenting the paper. We went through the comments and we answered in detail to all of them. An item-by-item reply follows for the revision.
The new manuscript has been modified according to the suggestions of the Reviewer. The parts modified are highlighted in red, while the new ones are highlighted in blue. The line numbers recalled in this document refer to the marked manuscript attached to the present document.

**Reviewer #1**
*In the manuscript "Extreme waves analysis based on atmospheric patterns classification:*
*an application along the Italian coast" the authors propose a methodology for classifying data of a physical quantity, to be applied prior to perform Extreme Value Analysis, for complying with three key requirements of the data samples: independent and identically distributed and, in addition for directional variables, grouped in homogeneous subsets. This last requirement is the principal objective of the manuscript, applied to significant wave height peaks along the Italian coast. Following previous works, they propose to use the atmospheric processes producing extreme wave conditions in a given location, (1) to select the homogeneous subset based on the weather patterns (WPs), and (2) to estimate the overall extreme values distribution starting from the distribution fitted to each subset.*
*The method relies on the physical connection between the atmospheric processes, spatial and temporal evolution of the surface pressure and wind fields, behind the occurrence of the extreme wave conditions at a given location. Consequently, the classification of extreme events is based on observed surface wind fields during the hours before and concomitant to the time of the peaks and on the correlation maps between the wind velocities and significant wave height peaks. Once the wind fields producing the peak wave conditions are identified the wind fields were used for clustering and classifying the extreme events. Then, the classification of the peaks, once the threshold of the wave height is chosen, depends only on the normalized wind fields.*
*The manuscript addresses relevant scientific and technical questions within the scope of NHESS, wave climate extreme value problem, presents new data and some novel concepts, with well-developed tools and very interesting results. Then it is worthy to be published in NHESS. However, before to recommend the manuscript for publication, there are, in my opinion, two important questions which the authors should clarify:*
*(a) The range of validity of the classification of extreme events, only based on surface wind fields.*
*(b) The "quality" of the obtained homogeneous data sets resulting from the feeding the k-means algorithm with the normalized wind fields producing the wave height peak conditions.*

As regards comment a), it should be pointed out that this study focuses on waves driven by wind; therefore, other physical quantities that may concur in generating sea waves (tides, soil vibrations etc.) are not considered. In case of wind waves, the variables that may affect their characteristics, beyond the wind, are the local bathymetry (geometry of the basin), sea level variations (i.e. water depth) and currents. However, in this case the bottom depth is not affecting the waves propagation, since all the locations investigated lie in deep water (see Table 1). As for the current, this was not considered in the wave generation model; nevertheless, the hindcast data were widely validated and proved to be reliable. Therefore, wind data are reasonable expected to be sufficient for the characterization of the extreme waves in our case. This has been strengthened at line 85. As such, the key point is how to define the time and spatial domain of the wind to be considered: since the Mediterranean Sea is an enclosed basin with

limited fetches, 12 hours were found to be enough; other areas may require different domains, depending on the characteristics of the basin.

As for comment b) (the "quality" of the subsets), the events belonging to different clusters are remarkably homogeneous in terms of wave bulk parameters, even though the latter were not used for clustering the peaks (this can also be appreciated with respect to Tp, as shown in the Figure at the end of this document). Moreover, in the framework of Extreme Value Analysis, data are homogeneous in term of the parent distribution, since thresholds were selected in order for the peaks to come from a Generalized Pareto Distribution by following the method proposed in Solari et al. (2017).

Finally, the homogeneity of the wave fields can be validated by looking at the mslp related to single events with respect to the overall mslp average. An example for location B7 is attached below, showing the mslp fields for the three highest events belonging to WP#2. These events were selected because they are characterized by Hs significantly higher than those of the point cloud, and are further recalled by the Reviewer in the next comment.

[Figure]

From the example it can be noticed how the mslp fields related to the single events are very similar between each other, and in turn to the average mslp field reported in Figure 4 (panel D) of the manuscript. The location of the low pressure of the cyclone is similar in both events, though there are differences in terms of the absolute value of the pressure and slightly in the shape of the cyclone too; something to be expected as it partially explains the different intensities of Hs (of course, local effects have to be taken into account as well).

**Developed Questions**

*The main difference between previous published research and present work is that the variable wind wave is generated under fetch and time limited conditions. Then, the correlation between observed wind velocity and wave height fields depends on the generation process quantified through the non-dimensional variables. Significant wave height and peak period, Hs\*, Tp\* depend on the nondimensional fetch, F\* and the non-dimensional time t\**

$$Hs^* = gHs/U^2 = f(gF/U^2, gt/U) \quad Tp^* = gTp/U^2 = f(gF/U^2, gt/U) \quad F^* = gF/U^2 \quad t^* = gt/U$$

*U is the mean wind velocity, at a certain height, over the fetch F. Fetch, and consequently U are defined based on the mean wind direction (well identified by the authors, figure 10).*
*These functional relationships between the non-dimensional quantities should be used to link the weather patterns, wind velocity and significant wave height time series. Based on that, F\* and t\* should be relevant quantities (bring in the physics of the wave generation process) to classify the extreme events and the correlation maps for different lags. For that purpose, the procedure defined by eqs (1) -(3) should be applied, not with wind velocity but with the non-dimensional quantities, and as well as to feed the k-means algorithm for finding the homogeneous subset.*

The relationships proposed by the Reviewer can be related to the so called Significant Wave Method. This model allows to predict the wave field at a given location by looking at non-dimensional ratios of the wind fetch (Bretschneider, 1959, among others), thus quantifying the wave generation process through non-dimensional variables. Although this theory is well known and established in the scientific literature, it refers to empirical formulations that may happen to be too simplified in case of random environmental conditions and complex coastline geometries.

Furthermore, the use of non-dimensional quantities introduces an additional problem: as shown by the correlation maps (Figure 2 of the manuscript), it is not possible to detect a prevailing fetch for the extremes at punctual location, and the wind velocities may vary dramatically along different fetches and time lags. Therefore, it would be difficult to define F and U to feed the non-dimensional relations with. On the other hand, this problem does not arise when raw wind data are used. Moreover, the wave model used to build the hindcast was fed with high-resolution wind data, allowing to describe in great detail the wave generation physic. As such, is Authors' belief that hindcast wind velocity data over the whole Mediterranean Sea provides the best information to be plugged into the k-means algorithm for classifying the Hs peaks.

As regard the homogeneity of the subsets, Reviewer is referred to the example reported in the previous comment: even though the highest two data points depart of approximately 15% from the third largest value of Hs, the mslp fields related to such events are very similar to the one corresponding to the third largest event (and to the average mslp field in turn, reported in Figure, 4 panel D).

*In addition, but very important issue, the non-dimensional quantities for each direction will help to define asymptotic values of the extreme distributions, as seems to occur figure 11. In location B4, it seems that there is an upper limit around Hs=8 m. This trend can be checked working with the non-dimensional quantities for bounded values of F and t for that direction. Similarly, for location B7, the two highest data points (Hs > 6m) depart approximately a 15% from the third largest value of Hs. Belong to the same subset of data? Please use the non-dimensional quantities for checking the homogeneity of the subsets.*

Regarding the definition of asymptotic values, we do agree in that some information could be extracted from the analysis of the dimensionless variables, but the task is not as straightforward as suggested by the Reviewer and, in our opinion, the value of the information obtained this way would not be exhaustive. As an example, we took buoy B7, as suggested by the Reviewer. For this buoy most extreme waves are associated with WP#2 (see Figure 10 in the manuscript) and come from WNW (approx. 290°). Dimensional fetch in this case is approx. 820 km (limited by the Balearic Islands); if wind speed of 22 m/s (number 9 in Beaufort scale) is considered and dimensionless relations between F and H are used as presented by Holthuijsen (2007), a value of Hs of approx. 9.9 m would be obtained. This is significantly larger than the observed values and is similar to the upper limit of the confidence interval shown in figure 11 of the manuscript.

As informative as this would be in terms of understanding the conditions generating an extreme event, there are many uncertainties in this kind of approximation that in our opinion prevent its use as an upper bound for the extreme distribution of Hs, namely: what wind speed should be considered? does it blow along the whole available fetch? and for how long? All these variables would significantly affect the computation of the Hs upper bound, and their definition would imply a high degree of subjectivity.

*Finally, while working with the wave height and neglecting the wave period, any risk analysis on the coastline or relevant maritime structures would be not complete. By using non-dimensional quantities, the values Hs and Tp are computed simultaneously because they depend on F\* and t\*. Please, if possible include the peak period to complete the necessary information for developing a risk analysis.*

We agree with the Reviewer that Tp is essential for developing risk analysis. Actually, classifying the wave fields according to the wind speed lead to clusters of Tp consistent with those of Hs, as the latter parameter is closely tied to the former one (especially in case of extreme sea states). This can be appreciated in the figure below, which shows the classification of Tp peaks according to θm for B4 and B7, similarly to what has been done in Figure 10 of the manuscript.

[Figure]

[Figure]

Once the series of Hs and Tp are defined, Extreme Value Analysis can be performed on the joint distribution of the two parameters (as an instance following Haver, 2015). This has been commented at line 301 of the manuscript.

[revised manuscript text omitted]

---

## Author Comment (AC2) · 20 Jan 2020

Dear Editor, we would like to thank the Reviewer for further reading and commenting the paper. We went through the comments and we answered in detail to all of them. An item-by-item reply follows for the revision. The new manuscript has been modified according to the suggestions of the Reviewer. The parts modified are highlighted in red, while the new ones are highlighted in blue. The line numbers recalled in the document refer to the marked manuscript attached to it.

Please also note the supplement to this comment:

[Figure]

https://www.nat-hazards-earth-syst-sci-discuss.net/nhess-2019-320/nhess-2019-320-AC2-supplement.pdf

---

## Author Response (AR1)

Dear Editor,
we would like to thank you for further reading and commenting the paper. We addressed all your comments. An item-by-item reply follows for the revision.
The new manuscript has been modified according to your suggestions. The parts modified are highlighted in red, while the new ones are highlighted in blue. The line and page numbers recalled in this document refer to the marked manuscript attached to the present document.

*- Consider writing a more effective abstract. The first three lines of the abstract can be dropped (they are not useful for appreciating the values of your study) and a short description of the methodology is not explicit.*

The abstract was rewritten dropping the first three lines and including a short description of the methodology.

*- At page 7," Algorithm 1 Computational scheme of empirical extreme values curves and its confidence intervals, has no clear status. Is it part of the text? Or it is a tale? I suggest to consider it a table (add a caption) and refer to it in the text. Further, the meaning of the individual lines with arrow and brackets is not clear. It needs to be substantially revised.*

The algorithm was modified in according to the Editor's suggestions, included into a table (Table 2 at page 8) and properly referred to in the text.

*- Figure 2 is not readable (arrows are too small)*

Figure 2 has been modified: the arrows width and length have been increased according to the suggestion.

*- The use of "unique" in the line 1 of the conclusions is ambiguous. "Results on the analysis of the extremes reveal how different weather circulation patterns may lead to sea storms having, on average, unique characteristics". As it stands, it might be interpreted that different weather patterns lead to the same marine storm*

We acknowledge that the terminology used in the above mentioned line may be misleading. The whole paragraph at hand has been rephrased to make it clearer and avoid misunderstanding.

*- In the paper, you talk about atmospheric processes, but I think you refer to circulation patterns (not processes). Please, correct the terminology*

Amended as suggested.

*- In the conclusion, when you write "For the analyzed locations, the proposed methodology led to the identification of two well-known cyclonic systems, characteristic of the atmospheric circulation on the Mediterranean Sea, as the origin of the extreme wave events affecting the Italian shores", I would suggest inserting a short description of those two patterns*

*- Further, you might discuss how your patterns compare with fig.122 of Lionello et al (2006). Cyclones in the Mediterranean region: climatology and effects on the environment. In P.Lionello, P.Malanotte-Rizzoli, R.Boscolo (eds) Mediterranean Climate Variability. Amsterdam: Elsevier (NETHERLANDS), 325-372, where the main synoptic circulation producing high waves in the Mediterranean are shown.*

Two paragraphs have been inserted in the manuscript according to your suggestions. The first at line 205, introducing a comparison with the work mentioned in the latter comment. The second at line 327, recalling the main characteristics of the circulation patterns identified.

*- The final sentences seem to suggest that your results are not really useful, because "The practical consequences of the classification in terms of the omni-WP extreme distribution are relatively limited…". Is this what you mean?*

What we really meant is that, even though the peaks are initially grouped in different clusters and analyzed separately, the long term distributions following the work-flow of Table 2 result to be consistent with the usual EVA scheme. The sentence has been rephrased to make it clearer and avoid possible misunderstandings.

*Further, I suggest you split section 3 in two new sections: "results" and "discussion" You may use the new section "discussion" to address the issues raised by the reviewers by using your comment from the interactive discussion. I refer to:*

- *Reviewer 1 who asks you to discuss a) The range of validity of the classification of extreme events, only based on surface wind fields. (b) The "quality" of the obtained homogeneous data sets resulting from the feeding the k-means algorithm with the normalized wind fields producing the wave height peak conditions.*

- *Reviewer 2 who claims that your proposed methodology for the classification of extreme events in the Mediterranean Sea may not be the proper approach to capture the characteristic wave climate, because of the, possibly high, local scale of the physical processes involved, especially when referring to extreme events. The table 1 and 2 that you added to the interactive discussion could be used for this answer.*

We acknowledge that the answers to the issues raised by the Reviewers could improve the overall quality of the manuscript. Therefore, these have been embedded in the text, and in particular in Chapter 3: "Results and Discussion".

However, we would prefer to keep Results and Discussions together, as in our opinion this facilitates to follow the work-flow. Indeed, ten Figures are presented, and to comment them all separately in a different chapter may result a bit confusing for many readers.

In particular, as for "The range of validity of the classification of extreme events, only based on surface wind fields", this was already commented at line 91, and has been further recalled at line 175.

Secondly, a new paragraph and two new figures (Fig. 5 and Fig. 6) have been added from line 211, in order to comment the homogeneity of the wave fields in term of mslp related to single events with respect to the overall mslp average, highlighting as well the divergences that take place (this is further recalled at line 236).

Furthermore, the whole paragraph between lines 277-287 has been rephrased, to highlight "how events belonging to different clusters are remarkably homogeneous in terms of wave bulk parameters, even though the latter were not used for clustering the peaks" (as previously commented in the first replies to the Reviewers).

Finally, a sentence has been added at line 300 to comment the intra-cluster homogeneity of the extreme events in term of their parent distribution.

[revised manuscript text omitted]

---

## Author Response (AR2)

02/04/2020

Dear Editor,
we would like to thank you again for further reading and commenting the paper. We addressed all your comments, and we modified the manuscript accordingly.
The parts modified are highlighted in red, while the new ones are highlighted in blue.

- We acknowledge that Figures 2, 3, 4, and 9 in their previous form were not clear, thus we reduced the number of arrows and increase their size and thickness.

- As far as the abstract is concerned, we modified the last paragraph according to the suggestion, but we did not add new text with the aim of complying with journal's recommendation of not exceeding 200 words in the abstract (we already have 210).

- As regard the last comment, we would like to point out that the analysis on the Weather Patterns (WPs) is not aimed at capturing local scale effects, nor it could allow to do that. Indeed, the characterization of the WPs is performed at large spatial scales (i.e. synoptic scale), starting from series of wave data defined at single locations. Possible local effects are accounted for in the wave numerical model employed to build the hindcast, as specified in lines 84-87.
  Starting from a series of data at a given location, we enlarge the overview on the synoptic scale effects that may be related to the observed wave peaks, but the characterization of the circulation patterns does not affect at all the selection of the peaks. For this reason, we talk of "bottom-up" scheme (line 52).
  Secondly, it is worth to mention that we do not claim that the proposed methodology yields an absolute advantage with respect to the usual Extreme Value Analysis (EVA) in terms of the return levels computation. On the contrary, in the text we draw the attention on the similarities we found, for our case study, between the return levels computed with the different approaches (Figure 13 and lines 282-288, 317-320), taken as granted the reliability of the common EVA scheme. However, this result is not to be generalized, as the difference between analyzing separately the different WPs populations and analyzing all the data pooled together would depends on the characteristics of the site (as pointed out at line 320). In any case, the methodology we propose allows to detail the circulation patterns most likely related to the most intense wave peaks at the investigated locations, which would not be possible without the initial clustering of the starting wave dataset. Moreover, the clustering result in homogeneous subsets of peaks, which can be therefore studied independently. As such, it is possible to compute the long-term return period quantiles through a Monte-Carlo scheme, and it is shown that no significant divergencies arise in the computation of the long-term return period quantiles, precisely.
  In view of the abovementioned considerations, we modified the Conclusions and reviewed the terminology to strengthen these points.

- We reviewed the text and fixed the spelling errors.

[revised manuscript text omitted]